# OpenReview forum: "SAIL: Self-improving Efficient Online Alignment of Large Language Models"
_ICLR.cc/2025/Conference — Submitted to ICLR 2025_

### Official Review · Reviewer_7i95 · 2024-10-24

**Soundness:** 3
**Presentation:** 2
**Contribution:** 3
**Rating:** 3
**Confidence:** 4

**Summary:**

The paper addresses the limitations of traditional reinforcement learning from human feedback (RLHF) methods for aligning large language models (LLMs) with human preferences. The authors propose a unified framework for online RLHF formulated as a bilevel optimization problem, which they simplify to a single-level method for efficiency. This approach, called SAIL, allows for continuous model improvement through online exploration and iterative refinement of preference labels, mitigating issues related to distribution shifts and reducing reliance on static preference oracles. Experimental results demonstrate significant performance gains, with SAIL outperforming state-of-the-art RLHF methods.

**Strengths:**

(1) The paper introduces a novel unified framework for online RLHF that effectively addresses the challenges of static datasets and distribution shifts.
(2) By reducing a bilevel optimization problem to a single-level method, SAIL maintains theoretical benefits while significantly lowering computational costs, making it more practical for real-world applications.
(3) The self-improving aspect of SAIL allows models to iteratively enhance alignment without extensive supervision, addressing the challenge of needing constant access to human preference data.
(4) Extensive experiments validate the effectiveness of SAIL, showing substantial improvements in performance metrics compared to existing methods, thus showcasing its applicability across various datasets.

I would consider rescoring if the authors can solve my concern.

**Weaknesses:**

(1) The method does not improve much in the AlpacaEval 2.0 Score. The author should give a detailed explanation. And why not use metrics like length-controlled win rate?
(2) Authors should compare more advanced preference optimization algorithms like ORPO and SimPO. And current results are not impressive for the alignment community.
(3) Why did the author just include MMLU as the downstream task metric? They should incorporate more tasks (eg., arc-challenge) like the similar self-improvement work SPIN (ICML24) to better illustrate their contribution.
(4) In the alignment area, it's better to conduct experiments in the Arena-Hard benchmark since it's a common metric to evaluate the alignment ability.

**Questions:**

See the weakness section.

---

> ### Author Response · Authors · 2024-11-23
> **Response Status Update to Reviewer 7i95**
>
> Thank you for your detailed comments. We are currently running the requested experiments and will post our complete responses with results soon. We appreciate your patience.

---

> ### Author Response · Authors · 2024-11-26
> **Response to Reviewer 7i95 (1/2)**
>
> > The method does not improve much in the AlpacaEval 2.0 Score. The author should give a detailed explanation. And why not use metrics like length-controlled win rate?
>
> **Response:** Thank you for your careful observation and question. We would like to clarify that we are already using the length-controlled (LC) AlpacaEval 2.0 win-rate metric in our evaluations. We will make this clearer in the table header of Table 3.
>
> Regarding the fact that the AlpacaEval 2.0 scores on LLama-3 (8B) do not improve compared to the baselines, we believe this is because our base model, the instruction-finetuned LLama-3 (8B), is already trained to perform exceptionally well in terms of helpfulness, which is the focus of the AlpacaEval benchmark. Additionally, the preference dataset we used, UltraFeedback, may not provide significant further enhancement in the helpfulness aspect. This is supported by the slight decrease observed in the AlpacaEval score for the standard DPO baseline as well (see Table 3, results on LLama-3). Therefore, we think these AlpacaEval 2.0 results on LLama-3 (8B) may not indicate that SAIL is ineffective; it may be simply caused by an ill-suited combination of base model, finetuning dataset, and evaluation benchmark.
>
> We also further conducted experiments on the Zephyr (7B) model as the backbone, whose AlpacaEval 2.0 win-rate is lower. We still train on the UltraFeedback preference dataset and the other experiment setups are unchanged. In this experiment, we see a larger improvement of the SAIL method compared to the standard DPO baseline (Zephyr-7B-Beta).
>
> |             | AlpacaEval 2.0 (LC) Win-Rate |
> |--------------------|------------------------------|
> | Base (Zephyr-7B-SFT-Full) | 6.4 %                        |
> | DPO (Zephyr-7B-Beta)   | 13.2 %                       |
> | SAIL-PP  | 15.9 %                       |
>
> > Authors should compare more advanced preference optimization algorithms like ORPO and SimPO. And current results are not impressive for the alignment community.
>
> **Response:** Thank you for raising this insightful point. We see ORPO and SimPO are two recent work which propose a different objective than the standard RLHF, and achieve remarkable improvements in terms of alignment performance and efficiency.
>
> Our work focus more on bringing standard RLHF to a bilevel optimization framework and propose an effective and efficient approximate algorithm on top of it. We can see some new preference optimization methods including ORPO and SimPO have one fundamental difference from our approach: they do not explicitly incorporate the KL regularization term. The absence of the KL regularization term allows these methods to optimize more aggressively for the reward function by deviating significantly from the reference model. In contrast, our approach is specifically grounded in the standard RLHF, where the KL regularization term ensures that the model remains aligned with the reference distribution while optimizing for the reward function. This distinction makes direct comparisons with ORPO or SimPO less meaningful theoretically, as those methods omit the KL regularization and adopt a fundamentally different optimization objective design.
>
> However, we think our work, although developed adhering to the standard RLHF setup, can be compatible and combined with some recent advanced preference optimization algorithms, despite their differences in optimization setups and objectives. This is because we can reformulate their alignment problem as bilevel optimization, and go through the derivation as done in the paper. Taking SimPO as an example, we can treat their reward model definition (Equation (4) in the SimPO paper) as the solution of the upper level optimization (replacing Equation (4) in our manuscript), and adopt their modified Bradley-Terry objective with reward margin (Equation (5) in the SimPO paper) to replace the standard one (Equation (10) in our manuscript). By applying these changes and rederiving the extra gradient terms, we can formulate an adaptation of our method to the SimPO objective. We will implement this combined algorithm, which adapt our methodology to the SimPO objective, and compare with the SimPO as a baseline.
>
> Recently many different alignment objectives and algorithms have emerged; it is an interesting question to discuss the compatibility and combination of our method with each objective. We will add more relevant discussions to the appendices, but due to the fact that the compatibility problem with each design is a non-trivial question, this process may incur considerably more work, and we hope the reviewer understands that this effort cannot be fully reflected by the rebuttal period. But we will continue to expand the discussion as the wide compatibility to other designs also strengthens our contribution to the community. We thank the reviewer for raising this insightful point.

---

> ### Author Response · Authors · 2024-11-26
> **Response to Reviewer 7i95 (2/2)**
>
> > Why did the author just include MMLU as the downstream task metric? They should incorporate more tasks (e.g., ARC-Challenge) like the similar self-improvement work SPIN (ICML24) to better illustrate their contribution.
>
> **Response:** Thank you for your suggestion. Let us first explain why we did not apply many different downstream task evaluation datasets to our experiments. One reason is that we have incorporated 5 other metrics including the widely used MT-Bench scores and AlpacaEval 2.0 length-controlled win-rates. Another reason is that the UltraFeedback fine-tuning dataset we used is primarily designed to consider 4 different aspects, namely instruction-following, truthfulness, honesty, and helpfulness, and therefore it may not be very useful to improve the model's capability on reasoning datasets like MMLU and ARC-Challenge.
>
> Nevertheless, we agree that adding more evaluation datasets would strengthen our experimental analysis. Following the reviewer's suggestion, we added the ARC-Challenge dataset as part of the evaluation and reconducted the experiments on Llama-3 (8B) and ARC-Challenge. We see similar observations as on MMLU, the SAIL methods bring larger improvements than the DPO baseline.
>
> |               | Instr-Tuned | DPO   | SAIL-PR | SAIL-PP | SAIL-DP |
> |---------------|-------------|-------|---------|---------|---------|
> | ARC-Challenge Accuracy | 82.2%       | 82.8% | 84.1%   | 83.6%   | 83.4%   |
>
> The results show that our improvements are larger than the DPO baseline, although the baseline improvement is small.
>
> > In the alignment area, it's better to conduct experiments in the Arena-Hard benchmark since it's a common metric to evaluate the alignment ability.
>
> **Response:** Thank you for your suggestion. We agree that the Arena-Hard benchmark is recently becoming a widely used benchmark. We use the Arena-Hard-Auto repository and adapt their newly introduced Style Control (SC) method, which follows an update of Chatbot Arena. Following the reviewer's suggestion, we added the Arena-Hard benchmark as part of the evaluation, and reconducted the experiments on Llama3 (8B). The observation is similar as on MT-Bench, where we clearly see the SAIL methods can lead to significantly larger improvements than the DPO baseline. We plan to add Arena-Hard evaluations to other experiments in the manuscript soon.
>
> |                                     | Instr-Tuned | DPO  | SAIL-PR | SAIL-PP | SAIL-DP |
> |-------------------------------------|-------------|------|---------|---------|---------|
> | Arena-Hard Score (Style Controlled) | 19.8        | 23.8 | 29.4    | 26.8    | 24.9    |

---

> > ### Author Response · Authors · 2024-11-28
> > **Looking Forward to Your Review of Our Responses**
> >
> > Thank you so much for your insightful and constructive feedback on our work. We have provided detailed responses to your valuable comments, including new experimental results on AlpacaEval 2.0 length-controlled win-rates with additional model architectures, as well as the requested Arena-Hard benchmark and ARC-Challenge evaluations.
> >
> > As we are nearing the end of the author-reviewer discussion period, we would be very grateful if you could take a moment to review our responses. We truly value your expertise and would welcome any additional thoughts or questions you may have. We are here to address any remaining concerns and continue this productive discussion.
> >
> > Thank you again for your time and dedication in helping us improve our work.

---

> > > ### Author Response · Authors · 2024-12-01
> > > **Time-Critical: Your Review of Our Responses Would Be Greatly Appreciated**
> > >
> > > We are nearing the end of the discussion period, and we wanted to reach out once more about our detailed responses to your insightful comments. We greatly value your thorough review and have worked diligently to address each of your concerns, including conducting additional experiments on AlpacaEval 2.0, Arena-Hard benchmark, and ARC-Challenge as per your suggestions.
> > >
> > > Your expertise and perspective have been crucial in strengthening our work, and we would deeply appreciate if you could take a moment to review our detailed responses before the discussion period ends.

---

### Official Review · Reviewer_ZoUS · 2024-11-01

**Soundness:** 4
**Presentation:** 4
**Contribution:** 3
**Rating:** 8
**Confidence:** 4

**Summary:**

The paper introduces SAIL (Self-improving Efficient Online Alignment), an approach for online reinforcement learning from human feedback (RLHF) that aims to align large language models (LLMs) with human preferences. SAIL addresses limitations in offline RLHF methods by framing online LLM alignment as a bilevel optimization problem, which it reduces to a single-level first-order optimization method to enhance computational efficiency. The approach allows for continuous model improvement by generating samples iteratively, regulating preferences, and exploring online feedback. SAIL's self-improvement mechanism enables it to reduce reliance on preference oracles, thus allowing for more scalable alignment. Empirical evaluations demonstrate significant performance improvements over standard RLHF baselines.

**Strengths:**

1. **Innovative Formulation**: The paper provides a novel formulation of online RLHF through bilevel optimization, enhancing computational efficiency by reducing this problem to a single-level optimization, which is a significant advancement for practical LLM training.
2. **Effective Self-improvement Mechanism**: SAIL effectively addresses challenges related to reliance on preference oracles, making online alignment more feasible by leveraging the model's self-generated responses for iterative improvement.
3. **Comprehensive Evaluation**: The paper includes extensive experiments that demonstrate substantial improvements in evaluation reward, win rate, and efficiency over other methods like DPO, supporting SAIL's efficacy and computational advantage.
4. **Scalability and Adaptability**: SAIL’s approach to handling distribution shifts and reducing oracle reliance presents a promising method for more scalable RLHF applications, especially for emerging large-scale LLMs.
5. **Detailed Experiment Design and Baselines**: The experiment section is well-structured, covering a range of metrics (reward-margin, eval-reward, win rate) and configurations (SAIL-PR, SAIL-PP, SAIL-DP), providing insights into the trade-offs and performance across different setups.

**Weaknesses:**

1. **Limited Exploration of Alternative Utility Functions**: The method relies on the Bradley-Terry preference model, which may not be optimal for all RLHF applications. Future work could benefit from exploring alternative utility models that account for more nuanced preference data.
2. **Scalability Concerns for Larger Models**: Although the paper demonstrates SAIL’s effectiveness on LLMs with up to 8B parameters, additional scaling experiments would strengthen the paper's claims about computational efficiency for significantly larger models.
3. **Dependency on Initial Offline Dataset**: While SAIL reduces oracle dependency, it still relies on an initial offline dataset to bootstrap alignment. Further discussion on managing this dependency, especially when starting with limited labeled data, could be beneficial.
4. **Potential Overfitting in SAIL-DP**: The paper mentions that SAIL-DP shows signs of overfitting on in-distribution responses, suggesting that the method may benefit from more refined regularization techniques to ensure robust generalization to out-of-distribution samples.

**Questions:**

1. The paper demonstrates SAIL's efficiency with models up to 8B parameters. Could you share any considerations or expected challenges for scaling SAIL to significantly larger models, such as those with over 100B parameters?

2. SAIL currently relies on the Bradley-Terry preference model. Have you considered experimenting with other preference models, and do you anticipate any impact on alignment performance if different utility functions are used?

3. SAIL-DP seems to show some overfitting on in-distribution responses. Could you discuss any regularization techniques you considered or plans to mitigate this, particularly to enhance generalization to out-of-distribution data?

4. Given the dependence on an initial offline dataset, how does SAIL perform in situations with minimal or noisy initial data? Are there strategies within the current framework to mitigate issues arising from a limited initial dataset?

5. Could you provide more detail on the computational costs of SAIL, particularly in comparison with other RLHF approaches? How does the single-level optimization approach compare in terms of resource requirements, and what practical considerations should be kept in mind when implementing it?

---

> ### Author Response · Authors · 2024-11-23
> **Response to Reviewer ZoUS (1/3)**
>
> > Limited Exploration of Alternative Utility Functions: The method relies on the Bradley-Terry preference model, which may not be optimal for all RLHF applications. Future work could benefit from exploring alternative utility models that account for more nuanced preference data.
> SAIL currently relies on the Bradley-Terry preference model. Have you considered experimenting with other preference models, and do you anticipate any impact on alignment performance if different utility functions are used?
>
> **Response:** Thank you for this insightful comment. Indeed, our current method relies on the Bradley-Terry (BT) preference model, and exploring alternative preference models is an exciting direction for future work. Since this is one of the initial works establishing a rigorous foundation for iterative RLHF, we focused on fundamental methods to clearly convey the core idea of Bilevel RLHF.
>
> Our work reveals a crucial insight: the BT preference model plays a critical role in ensuring strong concavity of the lower-level problem within our bilevel optimization framework. This mathematical property enables us to derive a closed-form solution, which is key to simplifying the bilevel problem into single-level optimization using the DPO trick. However, this approach may not readily extend to more complex or non-convex preference models, as they could introduce additional optimization challenges.
>
> We agree that extending the framework to accommodate alternative utility functions, particularly those capable of capturing more nuanced or domain-specific preferences, is a valuable research direction. Exploring these extensions could uncover interesting trade-offs between expressiveness, computational feasibility, and alignment performance, and we plan to address this in future work.
>
> > Scalability Concerns for Larger Models: Although the paper demonstrates SAIL’s effectiveness on LLMs with up to 8B parameters, additional scaling experiments would strengthen the paper's claims about computational efficiency for significantly larger models.
> The paper demonstrates SAIL's efficiency with models up to 8B parameters. Could you share any considerations or expected challenges for scaling SAIL to significantly larger models, such as those with over 100B parameters?
>
> **Response:** Thank you for this insightful question regarding the scalability of SAIL to larger models exceeding 100B parameters. We would like to share our considerations and expected challenges:
>
> 1.  **Primary Overhead Sources:** For the main SAIL methods—**SAIL-PP** and **SAIL-PR**—the major overhead compared to standard DPO comes from response generation and reward evaluation. The additional gradient terms computed (as per Equations (9) and (13)) are low-dimensional relative to the model parameters or inputs. This results in minimal time and memory overhead, even for models with over 100B parameters.
> 2.  **Challenges Similar to Online RLHF Training:** Scaling SAIL to larger models involves challenges common to most online RLHF training methods. To achieve computational efficiency and enable training on machines with limited resources, we recommend using **Parameter-Efficient Fine-Tuning (PEFT)** techniques not only for training but also during generation, as we have implemented in our code.
> 3.  **Technical Considerations:** There may be additional overhead when switching between training and generation modes, as well as interfacing with the reward model. Utilizing an optimized training framework that minimizes these overheads is crucial. Our current implementation adapts TRL's `DPOTrainer`, but it is not fully optimized or tested for models larger than 100B parameters. Further optimization is needed to handle the increased scale effectively.
>
> We believe that with these considerations and optimizations, SAIL can be effectively scaled to significantly larger models while maintaining computational efficiency.

---

> > ### Author Response · Authors · 2024-11-23
> > **Response to Reviewer ZoUS (2/3)**
> >
> > > Dependency on Initial Offline Dataset: While SAIL reduces oracle dependency, it still relies on an initial offline dataset to bootstrap alignment. Further discussion on managing this dependency, especially when starting with limited labeled data, could be beneficial.
> > Given the dependence on an initial offline dataset, how does SAIL perform in situations with minimal or noisy initial data? Are there strategies within the current framework to mitigate issues arising from a limited initial dataset?
> >
> > **Response:** Thank you for bringing up this important consideration. While SAIL does depend on an initial offline dataset to bootstrap alignment, it requires less initial data compared to standard DPO. This is because SAIL is designed to address the suboptimality issues of offline alignment methods and to be more efficient than exact bilevel formulations.
> >
> > In situations with minimal or noisy initial data, SAIL is better suited than standard DPO. Its reduced dependency on large amounts of high-quality data makes it more practical when starting with limited labeled data. Although mitigating issues from limited initial datasets isn't the primary motivation of our framework, this advantage allows SAIL to perform effectively even when the available data is minimal.
> >
> > > Potential Overfitting in SAIL-DP: The paper mentions that SAIL-DP shows signs of overfitting on in-distribution responses, suggesting that the method may benefit from more refined regularization techniques to ensure robust generalization to out-of-distribution samples.
> > SAIL-DP seems to show some overfitting on in-distribution responses. Could you discuss any regularization techniques you considered or plans to mitigate this, particularly to enhance generalization to out-of-distribution data?
> >
> > **Response:** Thank you for this insightful question. SAIL-DP does show signs of overfitting on in-distribution responses, as it significantly improves the Reward Margin but doesn't necessarily enhance metrics like the MT-Bench score. We hypothesize that this is due to the lack of exposure to out-of-distribution responses and offline rewards, which limits the model's ability to generalize.
> >
> > To mitigate this and enhance generalization to out-of-distribution data, we suggest the following strategies:
> >  - **Incorporate Out-of-Distribution Data:** Adding offline rewards and out-of-distribution responses to the training data can help the model learn a more generalized policy. This approach is employed in our SAIL-PR and SAIL-PP setups.
> >  - **Regularization Techniques:**
> >     - Data Augmentation: Augment the offline dataset by rewriting responses using other large language models (LLMs) to introduce more diversity.
> >     - Label Smoothing: Apply label smoothing techniques, such as those proposed in cDPO (Mitchell et al., 2023), to reduce overconfidence in the model and mitigate the impact of noisy preference labels.
> >
> > These strategies can help address the overfitting issue in SAIL-DP and improve its generalization to out-of-distribution samples.

---

> > > ### Author Response · Authors · 2024-11-23
> > > **Response to Reviewer ZoUS (3/3)**
> > >
> > > > Could you provide more detail on the computational costs of SAIL, particularly in comparison with other RLHF approaches? How does the single-level optimization approach compare in terms of resource requirements, and what practical considerations should be kept in mind when implementing it?
> > >
> > > **Response:** Thank you for your question regarding the computational costs of SAIL compared to other RLHF approaches. Here are our insights:
> > >
> > > 1.  **Overhead Comparison with Offline DPO:** SAIL introduces no additional overhead during the model update phase compared to offline DPO. The primary overhead stems from its online nature—specifically, response generation and reward evaluation.
> > > 2.  **Detailed Overheads of SAIL Variants:** As illustrated in Figure 5 of our paper, the overheads for the three SAIL setups vary:
> > >     -   **SAIL-DP:** This variant incurs minimal overhead, mainly from computing additional gradient terms during backpropagation.
> > >     -   **SAIL-PP:** In addition to the overhead in SAIL-DP, SAIL-PP includes significant overhead from generating online responses.
> > >     -   **SAIL-PR:** Beyond the overheads in SAIL-PP, SAIL-PR also involves overhead from reward evaluation.
> > >
> > >     By comparing the overheads of each setup, one can estimate the contribution of each component to the overall computational cost.
> > >
> > > 3.  **Resource Requirements and Practical Considerations:** Similar to other online RLHF methods, implementing SAIL requires careful management of memory resources due to the extra memory needed for online response generation and reward model evaluation. To optimize training speed, it's preferable to load all necessary models and caches into memory simultaneously to avoid the time overhead associated with frequent loading and unloading. Therefore, systems with larger memory capacity are advantageous for running SAIL efficiently.
> > > 4.  **Implementation Guidance:** Our code provides an example implementation based on the TRL package's `DPOTrainer`. While it may not represent state-of-the-art optimization, it serves as a practical starting point. Researchers can build upon this and explore additional optimization strategies to further reduce computational costs when applying SAIL to larger models.
> > >
> > > We hope this clarifies the computational considerations and practical aspects of implementing SAIL compared to other RLHF approaches.

---

> > > > ### Comment · Reviewer_ZoUS · 2024-12-02
> > > > **Response to rebuttal**
> > > >
> > > > Thank you for your response and addressing my concerns. I have no further questions and will keep my current rating.

---

### Official Review · Reviewer_urgR · 2024-11-03

**Soundness:** 3
**Presentation:** 4
**Contribution:** 3
**Rating:** 6
**Confidence:** 3

**Summary:**

The authors identify three significant challenges in online RLHF algorithms: Challenge 1: the interdependence between models and data in implicit reward learning; Challenge 2: the computational complexity of bi-level optimization; and Challenge 3: the reliance on preference oracles. They propose SAIL to address these challenges.

The main contributions of the paper can be summarized as follows:

1. **Unified LLM Alignment Mathematical Framework**: The authors have designed a principled online RLHF framework that provides concrete guidance for generating new responses, assuming the existence of a preference oracle.

2. **Adaptive Direct Preference Optimization**: By introducing a DPO-style analysis, the authors present an efficient single-layer solution capable of effectively addressing distribution shifts and providing a scalable online preference optimization method.

3. **Introduction of a Self-Improvement Mechanism**: This mechanism reduces the reliance on preference oracles.

4. **Extensive Experimental Evaluation**: The experiments conducted demonstrate that SAIL significantly outperforms baseline methods.

**Strengths:**

1. Introducing Bi-level Preference Optimization: The process of bi-level preference optimization is integrated into the modeling of online RLHF. By leveraging the unique correspondence between the reward function and the LLM policy, this approach innovatively transforms the process into an equivalent single-layer form that is easier to solve.

2. Extensive Experiments on SAIL: Comprehensive and rich experiments were conducted to address the three significant challenges in online RLHF and to demonstrate the relevant applications of SAIL.

**Weaknesses:**

Regarding the three variants of the SAIL method, Table 3 shows that in the Eval-Reward and MT-bench columns, the SAIL method performs worse than the baseline DPO. Please clarify whether these experimental results undermine the assertion that the SAIL method is superior to the baseline DPO.

**Questions:**

There is a large amount of blank space below Section 6.1. Is there any missing content in this part of the paper?

---

> ### Author Response · Authors · 2024-11-23
> **Response to Reviewer urgR (1/1)**
>
> > Regarding the three variants of the SAIL method, Table 3 shows that in the Eval-Reward and MT-bench columns, the SAIL method performs worse than the baseline DPO. Please clarify whether these experimental results undermine the assertion that the SAIL method is superior to the baseline DPO.
>
> **Response:** Thank you for your thorough analysis of our experimental results. In Table 3, we observe that among our variants, only SAIL-DP demonstrates marginally lower performance than the baseline DPO in Eval-Reward and MT-Bench metrics. However, this observation does not affect our broader conclusions regarding the effectiveness of our two primary SAIL implementations: SAIL-PR and SAIL-PP.
>
> Let us clarify the key points:
> - SAIL-DP employs a distinct methodology, utilizing responses from the offline dataset with self-generated preference labels. This contrasts with SAIL-PR and SAIL-PP, which generate responses online. Additionally, SAIL-DP operates with a reduced number of preference labels compared to standard DPO.
> - While SAIL-DP shows slightly decreased performance in Eval-Reward and MT-Bench metrics, it achieves notable improvements in Reward Margin. This is particularly significant given its reduced preference label requirements and minimal computational overhead.
>
> These findings support our overall conclusion regarding SAIL methods' superiority over baseline DPO. We will enhance the manuscript to better articulate the distinct characteristics and trade-offs of each SAIL variant.
>
>
> > There is a large amount of blank space below Section 6.1. Is there any missing content in this part of the paper?
>
> **Response:** Thank you for pointing this out. The blank space below Section 6.1 is not due to missing content; it is a LaTeX formatting problem. We will address this in the updated manuscript.

---

> > ### Comment · Reviewer_urgR · 2024-11-26
> >
> > Thank you for addressing my questions. I have no further inquiries and will maintain my current rating.

---

### Official Review · Reviewer_Rdtx · 2024-11-04

**Soundness:** 3
**Presentation:** 2
**Contribution:** 4
**Rating:** 6
**Confidence:** 4

**Summary:**

Compared to offline RLHF methods, online RLHF methods empirically show stronger performance, yet is computationally expensive, vulnerable to distribution shifts and lacks a unified framework. The authors ablate different online RLHF methods based on all possible combinations (namely, SAIL-PR, SAIL-PP, SAIL-DP) which could be useful for future work exploring online RLHF methods. Personally, it was surprising that SAIL-PP generally works on par or slightly better than SAIL-PR, which open up further research questions on what would be the optimal way to obtain preference dataset.

**Strengths:**

* The authors test of two LLM-as-a-Judge benchmarks as well as on a well-established classification benchmark, and their results are consistent.
* The authors provide a theoretical explanation of why their method works effectively.
* Showing all possible combinations at Figure 2 helped understanding what kind of online RLHF methods one should consider
* The results are consistent across smaller models (0.5B) up to widely used scale models (8B).

**Weaknesses:**

* As a practitioner, at least the presentation/writing wasn't clear enough to agree that SAIL provides a unified framework for those who might want to consider using online RLHF in future works. I would personally suggest adding a section explains about how one could use SAIL instead of iterative DPO methods, as well as a huge emphasis on how the provided code could be used.
* There is a huge emphasis on trying to improve reward models (on RewardBench) to mitigated reward model overoptimization & train better LMs. I am curious if given a fixed budget/time limit, whether one should try to employ online RLHF methods or try to enhance reward models in general.
* I would suggest adding an explanation of what is the limitation of online RLHF methods that the paper could not address. For example, it is still unclear on what is the best practice to "whether to discard instances from a preference dataset that have a subtle difference on the preference strength" or "would it be beneficial to employ more models when gathering responses when consisting a preference dataset".

**Questions:**

* Reward margin and offline-reward evaluation is interesting by itself and could provide information of the effectiveness of the method, but I personally think is not as an important measurement as pairwise winrate. Could you elaborate on Section 6.1 why one should consider looking into it?

* Please check the questions in weaknesses as well!

---

> ### Author Response · Authors · 2024-11-23
> **Response to Reviewer Rdtx (1/2)**
>
> > As a practitioner, at least the presentation/writing wasn't clear enough to agree that SAIL provides a unified framework for those who might want to consider using online RLHF in future works. I would personally suggest adding a section explains about how one could use SAIL instead of iterative DPO methods, as well as a huge emphasis on how the provided code could be used.
>
> **Response:** Thank you for this valuable suggestion. We will enhance the manuscript by adding a paragraph that addresses the limitations of current online iterative RLHF methods. In the final draft, we will expand upon the following points to better articulate the significance of SAIL:
>
>  - We will emphasize that iterative methods fail to account for interdependencies during the reward learning phase, specifically the dependency of policy-generated trajectories that result in distribution shift.
>  - To address these dependencies in a principled manner, we demonstrate the necessity of reformulating the alignment problem as a bilevel optimization problem, as expressed in equation (3).
>  - While bilevel optimization presents significant computational challenges due to its requirement for complex second-order information, making it computationally intensive.
>  - To overcome this, we leverage RLHF's special structure and the closed-form solution of the KL-regularized problem to transform it into a single-level problem without compromising generality, leading to our proposed SAIL approach.
>  - Finally, we develop a self-improvement mechanism that replaces the human-in-the-loop component by utilizing the implicit reward function as defined in equation (11).
>
> > There is a huge emphasis on trying to improve reward models (on RewardBench) to mitigated reward model overoptimization & train better LMs. I am curious if given a fixed budget/time limit, whether one should try to employ online RLHF methods or try to enhance reward models in general.
>
> **Response:** Thank you for raising this insightful point. Indeed, there has been significant emphasis on improving reward models (through initiatives like RewardBench and new VLM benchmarks, particularly from AllenAI etc.) which has successfully addressed certain issues such as length bias. While we acknowledge the value of this approach in addressing specific challenges, we believe the underlying issue is more fundamental and encompasses response quality more broadly. The effectiveness of reward models is intrinsically dependent on training with optimal or high-quality response pairs. However, this presents a significant challenge, as it necessitates training on an extensive corpus of responses to ensure comprehensive coverage.
>
> Our proposed bilevel optimization framework addresses this challenge by providing an efficient mechanism for concurrent training of the reward model and policy. This approach enables dynamic collection of task-relevant response pairs, resulting in more targeted and effective training.
>
> > I would suggest adding an explanation of what is the limitation of online RLHF methods that the paper could not address. For example, it is still unclear on what is the best practice to "whether to discard instances from a preference dataset that have a subtle difference on the preference strength" or "would it be beneficial to employ more models when gathering responses when consisting a preference dataset".
>
> **Response:** Thank you for this valuable suggestion regarding the limitations of online RLHF methods. We will include a comprehensive discussion of these limitations in the revised manuscript. Our theoretical insights and experimental analysis reveal an important finding: preference datasets containing diverse responses yield more informative gradients, which are essential for effective model updates. Conversely, responses with only subtle differences in preference strength generate minimal gradients, resulting in negligible improvements.
>
> Our work leaves several promising directions unexplored. One particularly intriguing possibility is the development of a curriculum-based approach that initially leverages diverse responses and progressively incorporates responses with closer preference values. Such an approach could optimize the learning process by capitalizing on response diversity in early stages while refining alignment as the model converges. This aligns with the natural progression we observe in model training, where response similarity tends to increase as the model approaches convergence, particularly in scenarios with low uncertainty in optimal response generation. This area represents a promising avenue for future research.

---

> > ### Author Response · Authors · 2024-11-23
> > **Response to Reviewer Rdtx (2/2)**
> >
> > > Reward margin and offline-reward evaluation is interesting by itself and could provide information of the effectiveness of the method, but I personally think is not as an important measurement as pairwise winrate. Could you elaborate on Section 6.1 why one should consider looking into it?
> >
> > **Response:** Thank you for this thoughtful feedback. While we agree that pairwise win rate represents a critical metric for response quality evaluation, reward margin and offline-reward evaluation contribute significant additional value for the following reasons:
> >  - These metrics enable quantitative comparisons between our method and baselines, demonstrating the effectiveness of our RLHF algorithm. Our evaluation utilizes high-quality offline reward models provided by the dataset authors, ensuring consistent evaluation standards.
> >  - Although we acknowledge the limitations inherent in using a static reward model, these metrics complement the pairwise win rate and other evaluations such as MT-Bench and MMLU. This multi-faceted approach provides a more comprehensive assessment of model performance.
> >
> > We think this combination of metrics offers a more complete understanding of our method's capabilities and limitations.

---

> > > ### Comment · Reviewer_Rdtx · 2024-11-25
> > >
> > > Thank you for the insightful responses. I will keep the current positive score as it is!

---

### Author Response · Authors · 2024-11-16
**Status Update on Additional Experiments**

Thank you for your detailed feedback and suggestions for additional experiments. We have carefully reviewed all comments and experimental requests, and are actively conducting the requested evaluations. We will provide a comprehensive response with results soon. We greatly appreciate your constructive feedback and patience as we work to strengthen our work.

---

> ### Author Response · Authors · 2024-11-23
> **Status Update on Response Progress**
>
> We have now posted detailed responses to questions that do not heavily depend on experimental validation. We are diligently working on the remaining experimental evaluations and will provide comprehensive results, along with any necessary response updates, in the coming days. We sincerely appreciate your thoughtful feedback and understanding as we work to thoroughly address all comments and strengthen our paper.

---

### Meta-Review · Area_Chair_uzDx · 2024-12-23

**Metareview:**

The paper introduces SAIL, a self-improving online RLHF approach for aligning large language models (LLMs). SAIL frames online alignment as a bilevel optimization problem, reducing it to a computationally efficient single-level method. The framework enables continuous improvement by iteratively generating samples, updating preference labels, and leveraging implicit reward functions. SAIL demonstrates performance gains on benchmarks like MT-Bench and RewardBench, reducing reliance on human preference oracles.

Key weaknesses include limited comparisons with recent methods (e.g., ORPO, SimPO), insufficient evaluation on diverse tasks, and unclear scalability and computational efficiency. The experimental setup lacks depth, with limited downstream tasks and inconsistent metrics. While the theoretical framing is novel, the practical contributions appear incremental.

**Additional Comments On Reviewer Discussion:**

Reviewers highlighted the novelty of framing online RLHF as bilevel optimization but raised concerns about evaluations, comparisons, and methodology. Authors provided additional benchmarks and clarifications, including Arena-Hard and ARC-Challenge experiments, but concerns about scalability, computational efficiency, and compatibility with other frameworks remained unresolved. Further comprehensive evaluations and theoretical discussions are needed.

---

### Decision · Program_Chairs · 2025-01-22

Reject